# Transport and Retention of Poly(Acrylic Acid-co-Maleic Acid) Coated Magnetite Nanoparticles in Porous Media: Effect of Input Concentration, Ionic Strength and Grain Size

**DOI:** 10.3390/nano12091536

**Published:** 2022-05-02

**Authors:** Rawan Mlih, Yan Liang, Miaoyue Zhang, Etelka Tombácz, Roland Bol, Erwin Klumpp

**Affiliations:** 1Institute of Bio- and Geosciences, Agrosphere (IBG–3), Research Centre Juelich (FZJ), 52425 Juelich, Germany; r.bol@fz-juelich.de (R.B.); e.klumpp@fz-juelich.de (E.K.); 2Institute for Environmental Research, Biology 5, RWTH Aachen University, 52074 Aachen, Germany; 3School of Resources, Environment and Materials, Guangxi University, Nanning 530004, China; liangyan@gxu.edu.cn; 4School of Environmental Science and Engineering, Sun Yat-sen University, Guangzhou 510006, China; zhangmy53@mail.sysu.edu.cn; 5Soós Ernő Water Technology Research and Development Center, University of Pannonia, H-8800 Nagykanizsa, Hungary; e.tombacz@chem.u-szeged.hu; 6School of Natural Sciences, Environment Centre Wales, Bangor University, Bangor LL57 2DG, UK

**Keywords:** coated magnetite nanoparticles, saturated column, breakthrough curve, deposition profile, mathematical modeling

## Abstract

Understanding the physicochemical factors affecting nanoparticle transport in porous media is critical for their environmental application. Water-saturated column experiments were conducted to investigate the effects of input concentration (Co), ionic strength (IS), and sand grain size on the transport of poly(acrylic acid-co-maleic acid) coated magnetite nanoparticles (PAM@MNP). Mass recoveries in the column effluent ranged from 45.2 to 99.3%. The highest relative retention of PAM@MNP was observed for the lowest Co. Smaller Co also resulted in higher relative retention (39.8%) when IS increased to 10 mM. However, relative retention became much less sensitive to solution IS as Co increased. The high mobility is attributed to the PAM coating provoking steric stability of PAM@MNP against homoaggregation. PAM@MNP retention was about 10-fold higher for smaller grain sizes, i.e., 240 µm and 350 µm versus 607 µm. The simulated maximum retained concentration on the solid phase (Smax) and retention rate coefficient (k1) increased with decreasing Co and grain sizes, reflecting higher retention rates at these parameters. The study revealed under various IS for the first time the high mobility premise of polymer-coated magnetite nanoparticles at realistic (<10 mg L^−1^) environmental concentrations, thereby highlighting an untapped potential for novel environmental PAM@MNP application usage.

## 1. Introduction

In recent years, a considerable expanding scientific interest in the fields of nanotechnology and nanoparticles has emerged due to the dramatic rise in their use. Iron oxide nanoparticles such as magnetite (Fe_3_O_4_), maghemite (γ-Fe_2_O_3_), and hematite (α-Fe_2_O_3_) have unique size-dependent properties which make them important for wide applications such as biomedicine [1], biosensors and drug delivery [2], catalysis [3], and environmental applications [4,5].

Iron oxide nanoparticles have been increasingly investigated for their clean-up potential of a wide range of contaminants [6], such as chlorinated organic compounds [7,8], pesticides [9,10], organic dyes [11], and heavy metals [12,13,14]. Magnetite nanoparticles (MNP) exhibit great potential for the removal of different types of pollutants from water [15,16,17,18]. The importance of these nanoparticles stems from their superparamagnetic properties and large specific surface area [19,20,21]. Additionally, MNP are economical, easy to synthesize, nontoxic, recyclable, and can be easily removed from the reaction medium once an external magnetic field is applied [22,23].

Applying surface coatings to bare MNP is highly desirable to improve colloidal stability as they have a positive effect on the surface charge status and aggregation potential [24,25]. Coatings can be derived from organic materials such as polymers, biomolecules, humic acid or from inorganic compounds such as silica, metals and metal oxides, or other materials [25,26,27,28,29]. 

Theoretically, the classical filtration theory (CFT) can be applied to describe the transport and retention of the nanoparticles in porous media. The attachment of small nanoparticles onto porous media surfaces occurs mainly by Brownian diffusion, while for larger particles, physical forces such as gravitational deposition and interception are the main mechanisms controlling their attachment [30,31]. Nanoparticles can be also deposited on porous media through a straining mechanism where nanoparticles are trapped in pore throats that are too small to let the particles pass [32,33]. The straining mechanism is highly dependent on the grain size of the porous media and colloid particle size [34]. Small grains and high surface roughness are believed to enhance colloid attachment [35,36,37]. Retained nanoparticles limit the available sites of the porous media, thus these sites become blocked or filled over time which stimulates the nanoparticle transport in the porous media [32]. 

Physicochemical properties of the collector such as surface roughness, chemical heterogeneity, and grain size, in addition to the other important factors such as solution pH, ionic strength, flow rate, and nanoparticle concentration, are critical for nanoparticle transport in porous media [38,39]. Ionic strength, for example, is a key factor controlling the mobility and deposition of iron oxide nanoparticles [40,41,42,43]. The increase in ionic strength results in electrostatic double-layer compression around the nanoparticles and the collector (the porous media grains), and the attractive forces that emerge consequently provoke nanoparticles aggregation and attachment to the collector and limit their mobility through the porous media [44,45]. Aggregation of iron oxide nanoparticles can also be triggered by increasing the concentration of the nanoparticles, which hinders their transport [1]. Furthermore, the aggregation can be promoted by magnetic attraction and pore clogging [27]. Therefore, understanding the transport behavior of MNP in porous media under these factors is essential for controlling their fate and application in remediation in the subsurface environment.

Coated MNP have been applied for a wide range of pollutants removal from aqueous solutions [46,47]. However, the transport of coated MNP in porous media or in soil systems has been less investigated [48,49]. The effect of initial concentration ranging from 100 to 600 mg L^−1^ on poly (acrylic acid) (PAA) coated magnetite in sand columns was previously investigated by [50]. The results showed that the mobility of the nanoparticles decreased as nanoparticle concentration decreased and the higher mass recovery at high concentrations was attributed to the time and concentration-dependent filling of retention sites. Additionally, high hydraulic gradients result in a high concentration of PAA-coated MNP in the effluent [51]. Particular studies have addressed the effect of different coating materials and surfactants on the transport of MNP in porous media to improve the stability and mobility of MNP in high saline oil reservoirs. These studies were oriented to the oil industry, and therefore they were carried out under extremely saline conditions (brine water, 1.9 M) and at very high nanoparticles concentration (2500 mg L^−1^). It was shown in a previous study [52] that the use of polymers and surfactants can enhance the mobility of coated MNP in porous media by 20%, with a recovery in the effluent of up to 97%. Higher retention of coated MNP was observed in Berea sandstone (d_50_ = 154 μm) due to high specific surface areas and chemical surface heterogeneity compared to Ottawa sand (d_50_ = 354 μm). Similar results were reported for nanoparticles coated by poly(2-acrylamido-2-methyl-1-propanesulfonic acid-co-acrylic acid) (poly(AMPS-co-AA)) under the same experimental conditions where high mobility (90%) was achieved in Ottawa sand while Berea sandstone fine grains retained ca. 41% of the nanoparticles [53]. Additionally, the use of (poly(AMPS-co-AA)) as a coating enhances MNP mobility by 96% at a concentration up to 1000 mg L^−1^ [54]. The high mobility was attributed to the coating effect that resulted in electrosteric stabilization of the nanoparticles and diminished the interactions of the nanoparticles with the negatively charged collector even at high salinity. Mathematical modeling based on a modified version of a multi-dimensional multispecies transport simulator (SEAWAT) supported the tracking of changes in electrolyte chemistry to predict its influence on the transport behavior of polymer-coated MNP in porous media [55,56]. 

The enhancement of coating the shell of MNP with poly(acrylic acid-co-maleic acid) (PAM) to improve the colloidal stability of MNP was firstly investigated for biomedical applications [24]. PAM showed superior abilities to be fastened to MNP in comparison with poly(acrylic acid) (PAA) because of the propensity of maleic acid to form metal−carboxylate complexes at oxide/electrolyte interfaces [57,58,59]. Additionally, PAM exhibits inner-sphere surface complex formation due to the high geometric matching between the carboxylate groups of maleic acid moieties and the surface sites of the crystalline phase of magnetite, whereas surface binding of PAA takes place only through H-bonds. PAM also showed high adsorption affinity and a very low concentration of free PAM in solution compared to PAA. Consequently, PAM adsorbed layer to a magnetite core exhibited high dilution resistance [60]. These arguments of colloidal physicochemical stability of PAM@MNP make them a superior candidate for environmental application. To the best of our knowledge, this study investigated for the first time the transport of PAM@MNP in porous media under varied parameters.

Previous studies explored the transport of the polymer-coated MNP at relatively high to extremely high concentrations and high IS and focused on rather small grain sizes. The transport of coated MNP in a low concentration range that is compatible with environmental standards and at low IS values similar to those applied in real conditions for water or soil remediation is hardly investigated. The objective of this study was thus to evaluate the effect of low initial concentration (1, 5, and 10 mg L^−1^) and IS (1, 5, 10, 50, and 100 mM) on the transport of PAM@MNP in quartz sand with grain sizes of 240, 350, and 607 µm. The experimental results were fitted using a numerical one-dimensional model of the advection–dispersion transport equation based on the classical colloid filtration theory to provide insights into the evolution of breakthrough curves (BTCs) and retention profiles (RPs) for PAM@MNP. 

## 2. Materials and Methods

### 2.1. Materials

Quartz sand (QS) with sizes of 607, 350, and 240 μm (Quarzwerke, Frechen, Germany) was used as a model porous media. Prior to column packing, the sand was purified to eliminate contamination with metal oxide contents that may interfere with nanoparticle transport; the washing step is explained elsewhere [61]. The Brunauer–Emmett–Teller (BET) measurement (Quantachrome, Syosset, NY, USA) for QS is 880, 380, and 50 cm^2^ g^−1^ For 240, 350, and 607 µm grain size, respectively. The zeta potential is −35.7 ± 3.4 mV at neutral pH conditions and IS of 1 mM, and Fe content is 0.002% [37].

The investigated nanoparticles, PAM@MNP, were developed at the University of Szeged in Hungary. In brief, synthetic magnetite (Fe_3_O_4_) nanoparticles were prepared by co-precipitation of Fe (II) and Fe (III) salts in an alkaline (NaOH) medium and purified by dialysis and magnetic separation. The product was coated with poly (acrylic acid)—PAA and poly (acrylic acid-co-maleic acid)—PAM [62].

The electrolyte solution, potassium chloride (KCl) (Sigma-Aldrich, Schnelldorf, Germany) with the desired ionic strengths of 1, 5, 10, 50, or 100 mM was prepared to flush the column before conducting the transport experiments and to prepare the PAM@MNP suspension and tracer solution. Deuterium oxide (0.5 M of D_2_O; Merck, Darmstadt, Germany) was used as a non-reactive tracer to compare the reactive transport of PAM@MNP in column transport experiments and to determine the hydraulic properties of the porous media. The zeta potential and the hydrodynamic radius for PAM@MNP at pH 8.5 were measured directly after preparation using a Zetasizer (Malvern Instruments GmbH, Herrenberg, Germany).

### 2.2. Fixed–Bed Column Setup

A stainless-steel column, 3 cm in inner diameter and 12 cm in length, was used for all experiments. A PTFE mesh with 200 μm openings was fitted at the openings of the column to support the porous media. The column was connected to a peristaltic pump (MCP V 5.10, Ismatec SA, Glattbrugg, Switzerland) with a three-way valve to control the type of liquid flow. The flow pulse was set in an upwards mode to obtain a steady flow and saturated state; a schematic diagram of the water-saturated column set-up is shown in Appendix A. The pH for all the experiments was adjusted to 8.5 using 0.1 M of NaOH and HCl solutions. A fraction collector (Merck, Darmstadt, Germany) was used to collect the effluent samples at the outlet of the column. The column was wet packed with the QS using a suction pump and vibrated to remove air bubbles and to attain an even distribution for the sand.

Before conducting each transport experiment, the column was flushed with a pulse of around 50 pore volumes (PV) of electrolyte solution. A summary of the column properties, nanoparticles, and sand characteristics is shown in Table 1. Then a pulse of 100 mL of D_2_O was injected into the column followed by a washing step by electrolyte solution under the same operating conditions; the transport of non-reactive tracer is described in detail elsewhere [37]. The concentration of the D_2_O in the effluent was quantified using high-performance liquid chromatography (D-7000 HPLC, High-Technologies Corporation, Japan) with a RI detector L-2490. The BTCs of the tracer were fitted using CXTFIT 2.1 code (STANMOD software, USDA) to determine the value of dispersivity (Table 1).

### 2.3. PAM@MNP Column Experiments

The experiments for the transport of PAM@MNP at initial concentrations of 1, 5, and 10 mg L^−1^ were conducted following the same procedure for the tracer experiments. Therefore, a pulse of 100 mL of PAM@MNP suspension was injected followed by an elution step with KCl electrolyte solution. The Fe in PAM@MNP solution was digested with 2% HNO_3_ and analyzed using an inductively coupled plasma mass spectrometer (ICP-MS, Agilent 7500 ce, Agilent Technologies, Inc., 71,034 Böblingen, Germany). At the end of the transport experiments, the sand in the packed column was excavated in a 1 cm thick increment. Two weights of 50 mg of dried and milled sand were digested with 0.25 g lithium borate mixture for approximately 30 min at 1000 °C in a muffle furnace. Each sample melt was dissolved in 30 mL HCl (5%) and diluted to 50 mL volume. For iron determination, two aliquots of the obtained sample solutions were extra diluted to the ratio of 1:10 and analyzed using ICP-OES. 

### 2.4. Mathematical Modeling 

Version 4.14 of the HYDRUS-1D computer code [63] was used to simulate the BTCs and retention profiles of PAM@MNP for column experiments conducted under various physicochemical conditions. The aqueous and solid-phase mass balance equations for PAM@MNP are given in this model as:(1)$$\frac{\partial \left(\theta \omega C\right)}{\partial t}=\frac{\partial }{\partial z}\;\left(\theta \omega D\frac{\partial C}{\partial Z}\right)-\frac{\partial \left(qC\right)}{\partial z}-\theta \omega \psi k_{1}C$$
(2)$$\frac{\partial (\rho_{b}S)}{\partial t}=\theta \omega \psi k_{1}C$$
where θω is the volumetric water content; *C* [Nc L^−3^, Nc, and L denote the number of NPs and units of length, respectively]; [*T*; T denotes units of time]; *z* [L] is the distance from the column inlet, *D* is the hydrodynamic dispersion coefficient [L^2^ T^−1^]; *q* is the Darcy water velocity [L T^−1^]; ψ  (-) is a dimensionless function to account for time and depth-dependent blocking; $k_{1}$ [T^−1^] is the first-order retention coefficient; $\rho_{b}$ is the soil bulk density [ML^−3^, where M and L denote units of mass and length, respectively]; and *S* [Nc M^−1^] is the solid phase of PAM@MNP concentration.

In Equation (1), the first and second terms on the right-hand side account for dispersive and advective transport of PAM@MNP, whereas the third term is used to describe PAM@MNP retention on the solid phase. The value of dispersion coefficient (D) was defined by fitting the BTCs of the conservative tracer using CXTFIT 2.1 code [63]. The following equation was applied to elucidate the distribution of retained PAM@MNP in the sand profiles: (3)$$\psi =\left(1-\frac{S}{S_{max}}\right){\left(\frac{d_{50+Z}}{d_{50}}\;\right)}^{-\beta }$$
where *d_50_* [L] is the median grain size, β (-) is an empirical parameter that controls the shape of the spatial distribution of retained NPs. *S_max_* [N_c_M^−1^] is the maximum solid-phase concentration of deposited PAM@MNP. Time-dependent blocking/filling of retention sites using a Langmuirian approach [64] is addressed in the first term on the right side of Equation (3), this term indicates that the retention profiles become more uniform with depth as *S* is approaching *S_max_* [65]. The second term on the right side of the same equation describes depth-dependent retention which indicates that the retention rate increases with depth. This term becomes equal to 1 when β equals 0, and thus an exponential distribution of retained PAM@MNP is predicted with depth. Conversely, when β > 0, the retention profile of PAM@MNP exhibits a uniform or hyper exponential shape (e.g., a higher deposition rate close to the column inlet). Based on the information presented in the literature and its relevance for different-sized sand grains, a value of β = 0.432 was assigned in the current study. However, this value did not adequately illustrate the observed depth-dependency in retention profile shape for PAM@MNP [66]. The bulk density of porous media, flow velocity, volumetric water content, and dispersivity were obtained from the applied experimental conditions and the fitting parameters of the nonreactive tracer. The model parameters (*k*_1_ and S_max_) for PAM@MNP transport were determined by simultaneously inverse fitting to experimental BTC and RP data using the Levenberg–Marquardt nonlinear least-squares optimization algorithm [67] in HYDRUS-1D computer code [63].

## 3. Results and Discussion

### 3.1. Characteristics of PAM@MNP 

The results of DLS measurements for freshly prepared suspensions of PAM@MNP (pH value fixed at 8.5 for all samples) shown in Table 1 reveal that the hydrodynamic diameter ranged from 119.3 ± 0.6 to 128.7 ± 1.5 nm, which is consistent with findings in previous studies for the same product [24,62]. The PAM@MNP size remained stable and almost in the same range at different values of IS from 1 to 100 mM, and thus potential aggregation under high IS can be neglected. The average zeta potential values indicated a net negative charge of the PAM@MNP under varied IS (1, 5, 10, 50, and 100 mM), which is attributed to coating [2]. The greatest negative value for zeta potential (−73.0 ± 1.8 mV) was recorded at 1 mM IS and generally became less negative with increasing IS due to charge screening. 

### 3.2. Effect of Initial Concentration on PAM@MNP Transport and Retention

The observed and fitted BTC of the conservative tracer showed a symmetrical shape, which confirms that the columns were functioning well and in good hydraulic condition (Appendix A). The results of STANMOD modeling showed high compatibility with the observed BTC versus fitted ones (R^2^ > 0.96%). Figure 1 shows the observed and simulated BTCs and RPs for PAM@MNP with input concentrations of 1, 5, and 10 mg L^−1^. The observed BTCs for PAM@MNP were plotted as the normalized effluent concentration (C_e_/C_o_) versus PV. When compared to the tracer, the BTCs for all concentrations of PAM@MNP started at about 0.7 PV and reached a plateau slightly behind 1 PV. The highest obtained (C_e_/C_o_) value of 0.94 was observed at C_o_ = 10 mg L^−1^, whilst the lowest value of 0.69 was obtained at 1 mg L^−1^ input concentration (Figure 1a). 

The increase in C_o_ resulted in an increase in PAM@MNP mass recovered in the effluent (Table 2). The results are in agreement with a previous study by Ersenkal et al. [50] who reported reduced recovery for coated MNP in the effluent with decreasing input concentration. In their study, the nanoparticle mass retained within the sand was close to 50% of the total mass recovered for the lowest concentration of 100 mg L^−1^. The BTCs for higher input concentrations (5 and 10 mg L^−1^) revealed a steeper shape compared to the lower concentration (1 mg L^−1^). The highest proportional retention of PAM@MNP as shown in Table 2 was observed for the lowest concentrations (C_o_ = 1 mg L^−1^), about fivefold compared to the highest concentration (C_o_ = 10 mg L^−1^). Previous studies showed that higher concentrations of nanoparticles can result in higher relative effluent concentrations, steeper effluent curves, and a lower deposition [61,68]. These trends may be attributed to blocking behavior that diminishes retention over time as limited retention sites become filled up rapidly at higher concentrations [32,53]. The retention profiles (RPs) plotted as the normalized solid-phase concentration (S/C_o_) versus distance from the column inlet varied significantly as the deposition of the PAM@MNP was not uniformly distributed (Figure 1b). Observations in previous studies showed that RP of coated nanoparticles can exhibit nonmonotonic shapes [28,68,69,70,71]. Liang [71] showed that the RP shape is altered by coating, resulting in nonmonotonic RPs that have peak concentrations at greater depths. The Hydrus-1 model was able to simulate the BTCs and the RPs (R^2^ > 0.9). The value of *k*_1_ and S_max_ both increased with decreasing *C_o_*_,_ which reflects higher retention rates and larger retention capacities at lower concentrations (Table 2). 

The model also predicted slightly higher retention of PAM@MNP at the column inlet, which may be attributed to the pore-scale hydrodynamic feature where the flux of nanoparticles solution is adjacent to the solid surface near the injection point [72]. Straining might be the mechanism behind the retention of PAM@MNP despite the small ratio of particle diameters to grain diameter (d_p_/d_c_), which is 2.06 × 10^−4^ in the case of the 607 µm sand grain. Hong et al. [73] reported that the straining d_p_/d_c_ ratio for different types of iron nanoparticles can be as low as 5.5 × 10^−5^. Similarly, Raychoudhury et al. [33] showed that straining can occur at very low ratios ranging from 2.24 × 10^−4^ to 1.23 × 10^−3^ for nZVI coated with carboxymethyl cellulose (CMC). 

### 3.3. Effect of Ionic Strength and Initial Concentration on PAM@MNP Transport and Retention

Figure 2 presents observed and simulated BTCs and RPs of PAM@MNP at different IS under C_o_ = 1 mg L^−1^ in 607 µm quartz sand. The BTCs under all the IS conditions showed a symmetric shape and the RPs varied extensively. The BTCs were moderately to well-described using the Hydrus-1 model (R^2^: 0.87–0.94), whereas the RPs were not accurately fitted. The PAM@MNP concentrations in the effluent increased when the IS decreased from 10 to 5 and 1 mM (Figure 2a). A corresponding decrease in RP took place, and the highest proportional retention rate in the sand (39.8%) was observed at 10 mM (Table 2). The *k*_1_ and S_max_ model parameters increased as the IS increased. The findings are in agreement with previous studies which showed that S_max_ and *k*_1_ decreased at low IS, as fewer sites are available for attachment, and thus blocking occurs faster than at higher IS [71,74].

The small input nanoparticle concentration resulted in higher relative retention when the IS increased. The same trend was found by Wang et al. [75], who demonstrated that smaller input concentration resulted in higher attachment efficiency at an increased IS for silica nanoparticles. The IS effect became less important as the input concentration of the nanoparticles increased. The BTCs and RPs obtained from 5 mg L^−1^ (observed BTCs and RPs are provided in Appendix A) show no significant influence in the range of 1–10 mM IS. Figure 3a shows steep and symmetrical BTCs even after high IS values (up to 100 mM) were applied at 10 mg L^−1^ initial concentration. The relative concentration (C/C_o_) of PAM@MNP in the effluent was beyond 0.8. These trends are different from a previous study that showed, at an initial concentration of 100 mg L^−1^, higher retention of PAA-coated MNP as IS increased to 10 mM [50]. The Hydrus-1 model could perfectly predict the observed BTCs (R^2^ 0.96–0.99). However, no obvious trend in *M_eff_*, *S_max_*, and *k_1_* values was observed (Table 2); this probably resulted from the high recoveries and the slight fluctuation in the column effluent under all the selected IS at this relatively high input concentration. 

Typically, the increase in IS for uncoated nanoparticles can result in particle aggregation due to the compression of the electrical double layer which allows attractive particle–particle interaction [76,77,78]. However, this effect is negligible in this case in the given IS range due to the electrosteric stability enhanced by the coating of the MNP [60,62], as also affirmed by DLS measurements (Table 1). The results are in agreement with previous findings [52,53] where high mobility of coated MNP under high saline conditions was observed, and the reduced amount of nanoparticles attached to sand and Berea sandstone surfaces was attributed to steric and electrostatic stabilization provoked by polymer layer on the surfaces of MNP. Similarly, Xue et al. [54] attributed the high effluent recoveries of coated MNP under high salinity conditions to colloidal stability as the nanoparticles remained negatively charged under high IS, and hence low attachment to silica grains occurred. 

The results obtained in our study are affirmed by Derjaguin–Landau–Verwey–Overbeek (DLVO) theory [30]. According to this theory, colloidal stability in a solution depends on the sum of van der Waals attractive forces and repulsive electrostatic forces. In the case of a polymer coating, steric repulsion also occurs [79]. In addition, if there are dissociable groups on the polymer chain, i.e., polyelectrolytes such as PAM here, the effects are combined, resulting in both steric and electrostatic, so-called electrosteric repulsion [24]. As the repulsive forces overcome the attractive forces, steric repulsions between the electrical double layers coated particles increase. In this case, a repulsive force could have emerged due to surface charge homogeneity of the PAM@MNP and the quartz sand as both possess negative charge at pH 8.5, which justifies the high concentrations of PAM@MNP in the effluent.

### 3.4. Effect of Grain Size on the Transport and Retention of PAM@MNP

The observed and simulated BTCs and RPs in Figure 4 indicate that the sand grain size had a strong influence on the transport and retention of the PAM@MNP. The model showed a good fitting for the observed BTCs (R^2^ > 0.90) and generally captured the shapes of the RPs. The observed transport and retention behavior of nanoparticles was reported by previous studies [71,80,81]. 

The proportional retention of PAM@MNP was 8 to 10 fold higher for the grain size QS of 240 µm and 350 µm, respectively, compared to 607 µm. Consistently, the fitted values of *k_1_* and *S_max_* increased with decreasing grain size, which means that the number of available sites for attachment of PAM@MNP increased as the surface area of the sand increased. The relatively low retention for 240 µm in comparison to 350 µm can be ascribed to the low total recovery for the 240 µm column (82.1%, Table 2).

The results are inconsistent with findings by previous studies for polymer-coated MNP transport in porous media, where greater mass breakthrough was observed for the coarse sand grains (d_50_ = 354 μm) compared to the fine sand grains (d_50_ = 154 μm) [52,53,56]. The high retention in finer grain sizes in these studies was attributed to the high specific surface area and chemical surface heterogeneity of the sandstone. 

Large-sized or coarsely textured media has been reported to limit the retention of nanoparticles and enhance their transport in the porous media due to the blocking effect, whereas fine grains of sand enhance higher retention as the blocking effect is less important [33,66,82,83]. Additionally, finer grain sizes are believed to enhance high pore diffusion, which in turn increases the rate of the mass transfer into the collector surfaces [34,84]. Furthermore, the roughness of the porous media and the presence of concave locations between roughness asperities could have also played a role in the higher retention of nanoparticles for these columns [36,85]. On the other hand, the increased retention of PAM@MNP with decreasing grain size can be attributed to the straining mechanism, which can occur at d_p_/d_c_ ratios as low as 0.0002. 

## 4. Conclusions

The transport of novel poly (acrylic acid -co-maleic acid) coated magnetite nanoparticles (PAM@MNP) was investigated using the column technique. The results showed that the transport of PAM@MNP was enhanced by an increase in input concentration (1–10 mg L^−1^) and grain size (240, 350, and 607 μm) of the porous media. The effect of high IS (100 mM) is important for PAM@MNP retention but only at low input concentrations. The high relative effluent concentrations achieved in most of the column experiments indicate that steric stability due to the coating effect diminished the attachment efficiency onto the sand grains and provided resistance against homoaggregation. The performance of PAM@MNP is attributed to PAM coating that has a high ability to be fastened on MNP due to the geometric matching between the carboxylate groups of maleic acid moieties and the surface sites of the crystalline phase of magnetite. The retention of PAM@MNP with decreasing sand grain size is believed to be attributed to the straining mechanism. The revealed high mobility of PAM@MNP is promising for environmental application and demonstrates the potential of using these nanoparticles for the removal of contaminants from soil and water.

## Figures and Tables

**Figure 1 nanomaterials-12-01536-f001:**
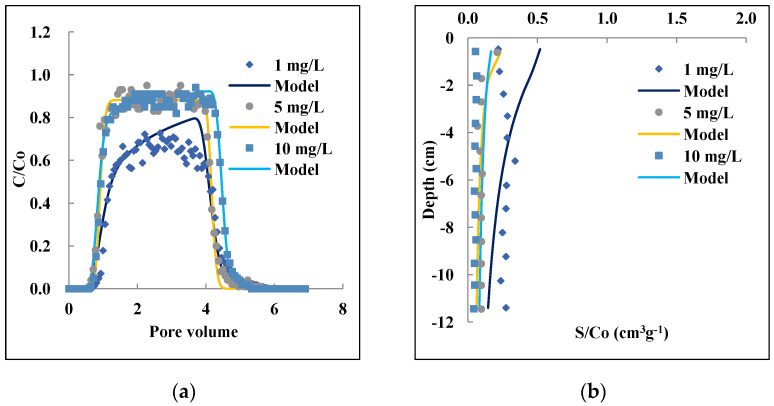
Effect of input concentration of 1, 5, and 10 mg L^−1^ on the transport and retention of PAM@MNP in saturated QS column: Observed and modeled BTCs (**a**) and RPs (**b**). IS: 1 mM, grain size: 607 μm, Darcy velocity: 0.28–0.29 cm min^−1^.

**Figure 2 nanomaterials-12-01536-f002:**
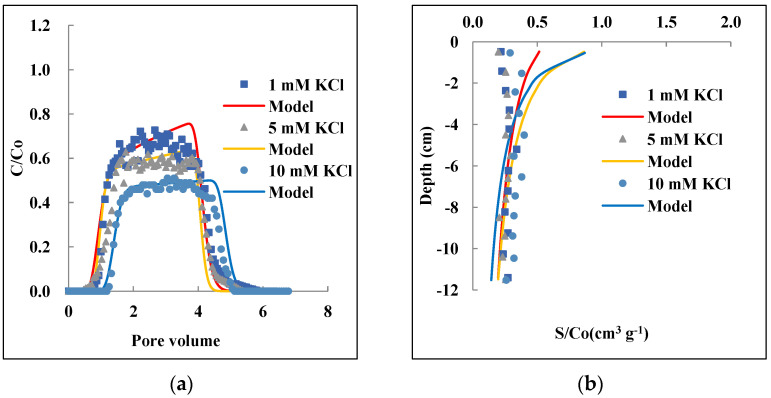
Effect of ionic strength 1, 5, and 10 mM on the transport and retention of PAM@MNP in saturated QS: Observed and modeled BTCs (**a**) and RPs (**b**). Initial concentration: 1 mg L^−1^; grain size: 607 μm, Darcy velocity: 0.29–3.0 cm min^−1^.

**Figure 3 nanomaterials-12-01536-f003:**
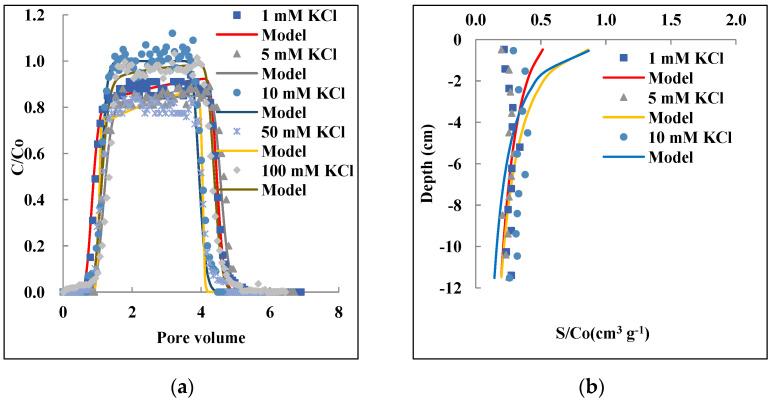
Effect of ionic strength 1, 5, 10, 50, and 100 mM on the transport and retention of PAM@MNP in saturated QS: Observed and modeled BTCs (**a**) and RPs (**b**). Initial concentration: 10 mg L^−1^; grain size: 607 μm, Darcy velocity: 0.29–3.0 cm min^−1^.

**Figure 4 nanomaterials-12-01536-f004:**
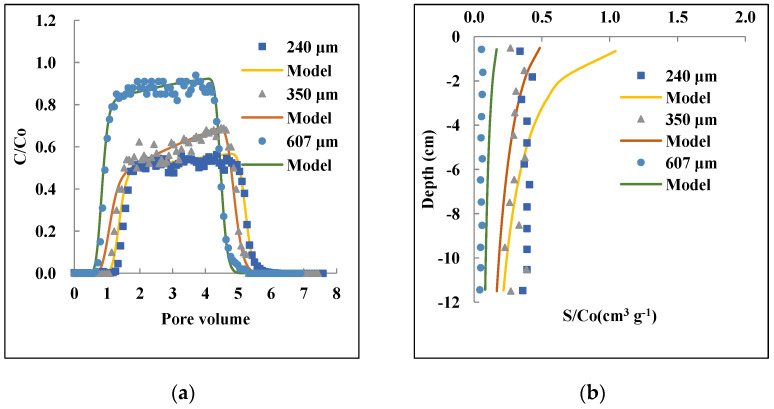
Effect of grain size 607, 350, and 240 µm on the transport and retention of PAM@MNP in saturated QS: Observed and modeled BTCs (**a**) and RPs (**b**). Initial concentration: 10 mg L^−1^, IS: 1 mM, Darcy velocity: 0.29–3.0 cm min^−1^.

**Table 1 nanomaterials-12-01536-t001:** PAM@MNP and QS characteristics and setup parameters for column experiments.

Figure	d_50_ (µm)	C_o_ (mg L^−1^)	IS (mM)	q (cm min^−1^)	φ	λ cm	ζ Potential of NP (mV)	Z_ave_-NP (nm)
1	607	1	1	0.29	0.38	0.146	−62.1 ± 3.4	126.2 ± 1.5
	607	5	1	0.28	0.34	0.171	−66.1 ± 2.7	128.7 ± 1.5
	607	10	1	0.29	0.34	0.175	−73.0 ± 1.8	121.0 ± 0.8
2	607	1	1	0.29	0.38	0.146	−62.1 ± 3.4	126.2 ± 1.5
	607	1	5	0.29	0.39	0.138	−59.1 ± 4.8	128.2 ± 3.0
	607	1	10	0.30	0.36	0.137	−27.1 ± 4.7	119.3 ± 0.6
3	607	10	1	0.29	0.34	0.175	−73.0 ± 1.8	121.0 ± 0.8
	607	10	5	0.30	0.36	0.154	−64.0 ± 1.4	122.1 ± 1.8
	607	10	10	0.28	0.39	0.153	−43.7 ± 2.1	121.6 ± 1.2
	607	10	50	0.30	0.43	0.953	−51.2 ± 3.6	115.4 ± 0.5
	607	10	100	0.30	0.38	1.887	−53.2 ± 2.3	115.5 ± 0.2
4	607	10	1	0.29	0.34	0.175	−73.0 ± 1.8	121.0 ± 0.8
	350	10	1	0.28	0.35	0.213	−73.0 ± 1.8	121.0 ± 0.8
	240	10	1	0.28	0.35	0.094	−73.0 ± 1.8	121.0 ± 0.8

d_50_: grain size; C_o_: input concentration of NP; IS: ionic strength; q: darcy velocity; φ porosity; λ: dispersivity (obtained by fitting tracer BTC); ζ: zeta potential; Z_ave_: average hydrodynamic diameter.

**Table 2 nanomaterials-12-01536-t002:** Experimental and model parameters and column experiments’ mass recovery.

Figure	C_o_ [mg L^−1^]	d_c_ [µm]	IS [mM]	q [cm min ^−1^]	*k*_1_ [min ^−1^]	SE *k*_1_	S_max_/C_o_ [cm^3^ g^−1^]	SE S_max_/C_o_	R^2^	Recovery %		
										M_eff_	M_sand_	M_total_
1	1	607	1	0.29	0.25	0.02	0.58	0.06	0.94	65.6	27.7	93.3
	5	607	1	0.28	0.08	0.00	0.27	0.03	0.99	84.3	18.2	95.9
	10	607	1	0.29	0.09	0.01	0.18	0.02	0.99	88.4	5.7	94.1
2	1	607	1	0.29	0.23	0.01	0.57	0.06	0.95	65.6	27.7	93.3
	1	607	5	0.29	0.23	0.01	1.63	0.27	0.90	58.1	26.8	84.8
	1	607	10	0.30	0.25	0.02	1.65	0.49	0.87	46.4	39.8	86.2
3	10	607	1	0.29	0.09	0.01	0.18	0.02	0.99	88.4	5.7	94.1
	10	607	5	0.30	0.06	0.01	0.71	0.58	0.98	85.7	4.8	90.6
	10	607	10	0.28	0.08	0.01	0.06	0.01	0.98	99.3	5.3	104.6
	10	607	50	0.30	0.11	0.01	0.35	0.07	0.96	79.4	5.3	84.7
	10	607	100	0.30	0.03	0.01	0.05	0.01	0.97	90.4	5.3	95.7
4	10	240	1	0.29	0.34	0.01	2.87	0.51	0.90	45.2	36.9	82.1
	10	350	1	0.28	0.38	0.03	0.48	0.05	0.94	56.7	45.5	102
	10	607	1	0.28	0.09	0.01	0.18	0.02	0.99	88.4	5.7	94.1

## Data Availability

Not applicable.

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
