# Peer review of "Transport and Retention of Poly(Acrylic Acid-co-Maleic Acid) Coated Magnetite Nanoparticles in Porous Media: Effect of Input Concentration, Ionic Strength and Grain Size"

_nanomaterials, 2022, doi:10.3390/nano12091536_

Round 1
Reviewer 1 Report
The Authors studied transport of a novel PAM-coated magnetite nanoparticles using column technique. They revealed that transport of nanoparticles is enhanced by increase of initial concentration from 1 to 10 mg and size of grains (240, 350 and 607 μm) which form the porous medium. The Authors found that effect of high ionic strength is important for the nanoparticle retention only at low concentrations. The high relative effluent concentrations achieved in the column experiments indicated that steric stability due to coating effect diminished the attachment efficiency onto the sand grains and provided resistance against homo-aggregation. The retention of nanoparticles with decreasing sand grain size could be connected to straining mechanism. High mobility of nanoparticles discovered in this study would be promising for environmental application and demonstrated the potential usage of PAM-coated magnetite nanoparticles for the removal of contaminants from soil and water. The manuscript is well organized and structured. The material presented is new and I recommend this manuscript for publication after minor revision. My comments are as follows:
1)The system of differential and algebraic equations (1)-(3) is not formally supplemented with boundary and initial conditions to formulate a boundary value problem. How were the solid curves drawn in Figureы 1-4?
2)It is necessary to clarify how the results obtained by Authors are affirmed by Derjaguin–Landau–Verwey– Overbeek (DLVO) theory. It is unclear from the text.
Author Response
"Please see the attachment"

Reviewer 2 Report
line 142: 250 must be changed by 350
line 164: superfluous word " schematic"
Author Response
Dear Ms. Miona Hu, Dear Reviewer
Thank you for giving us the opportunity to submit a revised draft of our manuscript titled “Transport and Retention of Poly(acrylic acid-co-maleic acid) Coated Magnetite Nanoparticles in Porous Media: Effect of Input Concentration, Ionic Strength and Grain Size to Nanomaterials journal.
We appreciate the time and effort that you have dedicated assess our manuscript. Changes were made according to the reviewer's suggestion:
Comments from reviewer 2.
line 142: 250 must be changed by 350
Response: Corrected.
line 164: superfluous word " schematic
Response: Corrected